# Ecuador Towards Zero Leprosy: A Twenty-Three-Year Retrospective Epidemiologic and Spatiotemporal Analysis of Leprosy in Ecuador

**DOI:** 10.3390/tropicalmed9100246

**Published:** 2024-10-19

**Authors:** Santiago Hernandez-Bojorge, Tatiana Gardellini, Jeegan Parikh, Neil Rupani, Benjamin Jacob, Ismael Hoare, Manuel Calvopiña, Ricardo Izurieta

**Affiliations:** 1Global Communicable Diseases, College of Public Health, University of South Florida, Tampa, FL 33620, USA; jeeganparikh@usf.edu (J.P.); bjacob1@usf.edu (B.J.); 2Office of Graduate Studies, Universidad Especializada de las Americas, Panama City 0833, Panama; tatiana.gardellini.1325@udelas.ac.pa; 3College of Medicine, University of South Florida, Tampa, FL 33620, USA; neilrupani@usf.edu; 4Global Health Practice, College of Public Health, University of South Florida, Tampa, FL 33620, USA; ihoare@usf.edu; 5One Health Research Group, Universidad de las Americas, Quito 170124, Ecuador; manuel.calvopina@udla.edu.ec; 6School of Public Health and Health Sciences, California State University, Dominguez Hills, Carson, CA 90747, USA

**Keywords:** epidemiology, spatial analysis, prevention and control, leprosy, Ecuador

## Abstract

Ecuador has gone through a significant reduction in new cases from 2000 (106) to 2023 (12), suggesting a trend towards zero leprosy. An ecological spatiotemporal study design was used to describe the epidemiological distribution of the disease in the country during 2000–2023. Leprosy cases registered by the surveillance system of the Ecuadorian Ministry of Public Health were the data utilized for the study. From January 2000 to December 2023, 1539, incidence cases were diagnosed with leprosy in Ecuador. At the time of diagnosis, the median age was 54 years. Most of the cases were males (71.5%). The proportion of incidence cases in subjects over 50 years was 63% and 1.5% in children ≤ 15 years old. The yearly incidence rate ranged from 8.5/1,000,000 population in 2000 to 0.68/1,000,000 population in 2023, remaining within the low-endemic parameter. In total, 35 cantons reported newly detected leprosy cases in the year 2000. By the end of 2023, only eight cantons actively reported cases of leprosy. High-risk clusters for leprosy were detected in the tropical coastal region of Ecuador. The provinces with the highest number of cases during the study period were Guayas (44.8%) and Los Rios (15.7%), with zero cases being found in the Galapagos Islands. Our study is unique in that it documents a retrospective dataset over a two-decade timespan from a South American country that has effectively applied global guidelines for the control and elimination of leprosy.

## 1. Introduction

Leprosy is classified as a neglected tropical disease (NTD) and is reported to have one of the highest numbers of incidence cases reported annually when compared to other NTDs globally (e.g., dracunculiasis, Buruli ulcer, Human African Trypanosomiasis (HAT), Visceral Leishmaniasis (VL), yaws; only Chagas’ Disease reports higher annual incidence cases) [1]. Leprosy, also known as Hansen’s disease, is characterized as a chronic infectious disease caused by the slow-growing acid-fast bacilli *Mycobacterium leprae* and *Mycobacterium lepromatosis* [1,2]. Transmission occurs through respiratory droplets released by coughing or sneezing during prolonged close contact with untreated individuals, with the mycobacteria showing tropism for macrophages in the skin, the mucosal lining of the respiratory tract, the eyes, and the Schwann cells of peripheral nerves [3]. The microorganism is slow-growing, with the incubation period ranging from 2 to 12 years, with an estimated average of five years [3]. Variations in the incubation period depend on the clinical form of the disease, appearing shorter for paucibacillary (PB) disease (2–5 years) compared to multibacillary (MB) disease (5–10 years, or longer) [4]. Leprosy was classified by Ridley and Jopling based on histological and immunological features into five types: tuberculoid (TT), borderline tuberculoid (BT), mid borderline (BB), borderline lepromatous (BL), and lepromatous leprosy (LL) [5]. The ICD-10-CM Code for Leprosy [Hansen’s disease] is A30. However, for treatment purposes, leprosy is classified as paucibacillary (PB) or multibacillary (MB) based on the number of skin lesions, the presence of nerve involvement, and the identification of bacilli on slit-skin smear. PB leprosy cases are characterized by 1–5 skin lesions, without the demonstrated presence of bacilli in the skin smear. In contrast, MB cases are identified with more than five skin lesions; nerve involvement (pure neuritis, or any number of skin lesions and neuritis); or the demonstrated presence of bacilli in a slit-skin smear irrespective of the number of lesions [6]. Therefore, indeterminate, TT, and BT cases can be included in the PB group, while BB, BL, and LL patients can be classified under the MB group [7]. This chronic infectious disease is curable, and disability can be prevented if early treatment is provided. The World Health Organization (WHO) recommends multidrug therapy (MDT) consisting of dapsone, rifampicin, and clofazimine for 6 or 12 months for treating PB or MB cases, respectively [7]. The WHO guidelines for the diagnosis, treatment, and prevention of leprosy are followed by Ecuador [7].

Since the recommendation of MDT in 1981 by the WHO, the worldwide prevalence of leprosy has decreased significantly, with approximately 16 million leprosy patients being treated with MDT in the last 20 years [8]. In 2020, 127,558 new leprosy cases were detected worldwide with a prevalence rate of 16.7 per million population. The case detection rate of leprosy in children below 15 years of age was 4.4 per million population, with the total number of cases being 8629 [8]. However, despite the decrease in case notification rates, it has been suggested by epidemiological and spatiotemporal studies conducted in Brazil and China that leprosy has focal distribution and the potential for hidden cases, necessitating intensified targeted control activities to reach the goal of the ‘Towards Zero Leprosy’ strategy [9,10,11,12]. In a study conducted in the Philippines in 2013, static, low transmission of leprosy in endemic rural areas and among children was identified despite good MDT coverage, and a new approach of chemoprophylaxis/immunoprophylaxis was suggested to target leprosy elimination [13]. Social vulnerability factors, such as poverty, food shortages, and associated nutritional deficiencies, as well as demographic factors, such as being male and of an older age, are the risk factors associated with leprosy [9,14,15,16,17]. 

In 2020, the highest proportion of cases was detected in the Southeast Asia region, followed by the Americas, with 74% of all worldwide new cases being reported by Brazil, India, and Indonesia [8]. A new case detection rate of 18.8 per million population was recorded in the Americas, which is higher than the worldwide new case detection rate of 16.4 per million population. In 2020, the number of new cases detected in Ecuador was 14, with a registered prevalence of four at the end of 2020, which is nearly double the prevalence at the end of 2019 [8]. The detection of new cases in several provinces and cantons of Ecuador suggests active transmission of the disease in specific geographic areas of the country. Overall, Ecuador has gone through a significant reduction in annual incidence from 2000 (106) to 2023 (12), suggesting a trend towards zero leprosy. Still, in the last 10 years, there have been increased cases in 2015, 2019, and 2021, which may be explained by more intense surveillance and detection of new cases. 

In April 2021, the WHO released a new ‘Towards Zero Leprosy’ strategy in line with the NTDs road map to break the chain of transmission and eliminate leprosy [2,8]. Specific targets under this strategy include the reducing of autochthonous cases to zero in 120 countries, a 70% reduction in the annual number of new cases, and a 90% reduction in the rate per million of new child cases [2]. A leprosy notification system called SIVE-ALERTA (Sistema Integrado de Vigilancia Epidemiológica/Integrated System of Epidemiologic Surveillance) is used by the Ecuadorian Ministry of Public Health (MSP) through an electronic PAHO-supported platform called ViEpi (Vigilancia Epidemiológica). Currently, new cases are reported through the SIVE-ALERTA system using both passive and active notification strategies. This ecological spatiotemporal study was conducted using official leprosy case data from 2000 to 2023 to further understand the epidemiology of leprosy in Ecuador and update the knowledge related to the status of leprosy in the country. Focal areas with transmission issues and vulnerable populations will also be identified, and these can be specifically targeted to achieve the goal of zero autochthonous cases in line with the WHO’s “Towards Zero Leprosy’ strategy”.

## 2. Materials and Methods

### 2.1. Study Area

Ecuador is located on the Pacific coast in the northwestern part of South America. Its unique and heterogeneous ecology is determined by the presence of the Andean mountains, which divide the country into three continental habitats: Costa (Pacific coast), Sierra (Andes), and the Amazonian regions. Additionally, the country encompasses one of the most important niches of aquatic fauna and flora in the world: the insular Galapagos Islands in the Pacific Ocean. Distinct ecological diversity is found in each region (Figure 1). The Costa (Pacific coast) consists of fertile tropical plains, hills, and low-elevation regions located in the western portion of the country, including the Pacific coastline. Guayaquil (Ecuador’s largest city and main port city), Esmeraldas, and Manta are some examples of major cities located in the warm and humid Costa. The Sierra (Andes) region is in the central belt of Ecuador, extending itself from north to south, including the elevated Andes mountains and the volcano chain. This region is well known for its agricultural capacities, its national parks, and major cities, such as Quito (the country’s capital), Cuenca, and Riobamba. The Amazonian region extends from the eastern slopes of the Andes into the plains of the Amazonian forests. This region includes biosphere reserves and indigenous middle-size towns, such as Tena, Coca, and Lago Agrio. The Galapagos Islands are also known as the “Archipiélago de Colón” and consist of 13 main volcanic islands and 17 islets located about 1000 km west of the mainland Pacific coast [18]. The country is divided into 24 provinces (7 on the coast, 10 on the Andes, 6 on the Amazon, and 1 in the Galapagos Islands) which are sub-divided into 224 cantons [18].

### 2.2. Study Subjects

Confirmed leprosy cases registered by the surveillance system of the Ecuadorian Ministry of Public Health (“Ministerio de Salud Pública” (MSP)) and the Ecuadorian Agency of Statistics (“Instituto Nacional de Estadísticas y Censos” (INEC)) according to the international codification (ICD-10 A30.1-9) were the data utilized for the study. Leprosy is a national notifiable disease in Ecuador, and all health personnel (public or private) are obligated to report suspected and confirmed cases. SIVE.ALERTA is part of the “Dirección Nacional de Vigilancia Epidemiológica” (National Directorate of Epidemiological Surveillance). The Epidemiological Gazette provides timely national information generated by the operational establishments of the Public Health System and Complementary Networks. This information is collected from the SIVE-ALERTA surveillance subsystem, which monitors events with high epidemic potential, outbreaks, and endemic diseases (publicly available at https://www.salud.gob.ec/gaceta-indicadores-2024/ (accessed on 6 January 2024)). Therefore, all suspected and posteriorly confirmed leprosy cases are registered in SIVE-ALERTA (a subsystem of the Ministry of Public Health of Ecuador) and, later, in the INEC. In recent years, the MSP’s passive and active notification strategies were used to report new cases through the SIVE-ALERTA system. To maintain the most reliable leprosy incidence report nationwide, incidence cases from two official sources of the Ecuadorian government (INEC and SIVE-ALERTA) were obtained and analyzed. INEC datasets include the National Archive of Data and Statistical Metadata (Archivo Nacional de Datos y Metadatos Estadísticos—ANDA) of the Ecuadorian National Institute of Statistics and Censuses, which registry all deaths and hospital discharges (including leprosy cases) reported at the national level by the General Direction of Civil Registry and the Ecuadorian Ministry of Health. The INEC provides the official government website for ANDA, available at https://anda.inec.gob.ec/anda/index.php/catalog (accessed on 10 January 2024). SIVE-ALERTA (Epidemiological Surveillance System) collects cases through surveillance of outpatients diagnosed with leprosy attending a health unit, with later confirmation being obtained using laboratory testing or by epidemiological linkage.

The diagnostic criteria were based on clinical, microbiological, and histopathological features described by international and national clinical guidelines [6,19]. The paucibacillary (PB) or multibacillary (MB) classification was based on the number of skin lesions, the presence of nerve involvement, and the identification of bacilli on slit-skin smear. PB leprosy cases were characterized by 1–5 skin lesions without the demonstrated presence of bacilli on the skin smear. MB cases were characterized as having over 5 skin lesions; nerve involvement; or the demonstrated presence of bacilli on a slit-skin smear irrespective of the number of lesions [6]. The data on the total population size were collected from the World Bank data from Ecuador and the Ecuadorian Agency of Statistics (INEC) [20]. Basic demographic and clinical information was obtained for all patients (age, gender, ICD clinical classification, status of medical care-outpatient/hospitalized, and date of diagnosis). Age was categorized into four groups (<15, 15–39, 40–69, and >70) based on the age distribution of the Ecuadorian population and the incidence of leprosy cases among these groups [21].

The data from various sources were entered into the Excel worksheet and were used for further analysis. A descriptive statistical analysis was conducted using R (4.0.3) and a spatial analysis was conducted using ArcGIS Pro Software Version 3.3.0 (Environmental Systems Research Institute, Inc., Redlands, CA, USA). Microsoft Excel 2021 for Windows (Microsoft Office, Redmond, WA, USA) was used to create various charts. 

### 2.3. Statistical Analysis

The leprosy incidence case data from 2000 to 2023 were stratified based on sex, age groups, geographical distribution (province, regions), and leprosy ICD-10 classification. Descriptive statistics such as frequency (%) for categorical variables and median (interquartile range—IQR) for continuous variables were calculated. This was followed by calculating the annual incidence rate per 1 million population and the mean (sd) for cases/year. Age in years was transformed into a categorical variable (<15, 15–39, 40–69, and 70 and above). Crude population data for cantons/provinces/regions were used for the years in which the INEC developed its official national census (2001, 2010, and 2022), and mean populations were extrapolated for the remaining study periods (2002–2009, 2011–2021, and 2023). Crude incidence rates were then calculated for the entire study period (2000–2023). The number of cases per year and the annual incidence rate per million population were plotted on a mixed bar-line graph to determine the trend in leprosy cases during the study period. A univariate analysis (one-way ANOVA and chi-square tests/Fisher–Freeman–Halton exact tests) was developed to explore the relationship between the sociodemographic/disease status classification (number of cases) and the administrative regions of Ecuador. 

### 2.4. Spatial and Spatiotemporal Analysis

Spatial analysis of the cumulative cases from 2000 to 2023 per canton, province, and region was conducted using ArcGIS Pro Software Version 3.3.0 (Environmental Systems Research Institute, Inc., Redlands, CA, USA). Reporting of data from 2000 to 2023 was utilized to calculate long-term trends. The burden of the disease was measured through the Leprosy Burden Scale (LBS) developed by the WHO Regional Office for Africa [22]. This endemicity classification scale presents three main indicators: low endemicity level (New Case Detection Rate—NCDR < 10), medium endemicity level (NCDR = 10–20), and high endemicity level (NCDR = <10) [22]. Leprosy endemicity was measured by the new case detection rate (NCDR) and new cases among children. Leprosy burden refers to the broader concept of the impact of the disease in a community and includes NCDR, MB cases, cases in children/women, societal consequences of the persons living with the disability (social stigma), and prevalence. To identify possible areas of autocorrelation, the Anselin Local Moran’s I method (Local Indicators of Spatial Association—LISA) was used. This technique can identify spatial clusters of features with scattered values and spatial outliers by calculating the Moran’s I value, a z-score, a pseudo-*p*-value, and a code representing four possible high/low cluster types for each statistically significant feature (High–High cluster, High–Low outlier, Low–High outlier, and Low–Low cluster) [23]. For spatial representation, we used Moran’s maps, considering cantons with a statistically significant difference. Additionally, a hot spot analysis was developed to calculate the Getis-Ord Gi* statistic for each feature in the data. This technique reports z-scores and *p*-values for each feature in the dataset. Statistically significant hot spots present high values and are surrounded by other features with high values as well. The local sum for a feature and its neighbors is compared proportionally to the sum of all features; when the local sum is very different from the expected local sum, and the difference is too large to be a result of a random chance, a statistically significant z-score is reported. This analysis assumes that a high value of z-scores with small *p*-values in terms of a parameter indicates the spatial agglomeration of high values and that a low negative z-score with a small *p*-value indicates the spatial grouping of low values [24].

### 2.5. Ethics Statement

The INEC (Instituto Nacional de Estadísticas y Censos) official database and the Epidemiological Surveillance System (SIVE-ALERTA) system provided by the Ecuadorian Ministry of Public Health (MSP) is anonymized and publicly available [22]. Therefore, the datasets used in this study did not report personally identifiable information, such as name, national identification number, date/place of birth, or biometric records. This article reviews INEC and SIVE-ALERTA data and does not require bioethical approval. 

## 3. Results

### 3.1. Study Population

From January 2000 to December 2023, a total of 1539 incidence cases were diagnosed with leprosy in Ecuador. At the time of diagnosis, the mean age was 54 years (sd = 18) (Table 1). Most of the cases were males (71.5%) (Figure 1). Among the new cases of leprosy detected, the percentage of male patients ranged from 67% in the year 2000 to 92% in the year 2023. The proportion of incidence cases in subjects over 70 years was 23% (612) and 1.9% (30) in children ≤ 15 years old (Table 1, Figure 2). Crude numbers of leprosy cases were significantly higher among the older age groups when compared to the <15-year-old group and among patients with paucibacillary (PB) infections when compared to multibacillary infections (MB) (Table 1). There was no statistically significant relationship among the other sociodemographic–clinical variables, such as gender (*p*-value: 0.379) and status (*p*-value: 0.585).

The percentage of children under 15 years fluctuated from 0% in the year 2000 to 5.2% in the year 2005 and then back to 0% after 2019 (Table 2). Using the International Classification of Diseases (ICD-A30), 7.5% of incidence cases were lepromatous (116/1539), 0.3% were borderline lepromatous (4/1539), 0.8% were borderline (12/1539), 16.2% were indeterminate (249/1539), 0.9% were tuberculoid (14/1539) and 73% were unspecified cases (1123/1539). According to the WHO classification, 92% of the cases (1419/1539) were paucibacillary while only 8% were multibacillary (120/1539) (Table 1). One hundred and four of the cases (86.7%) were detected in the Sierra region. Among the new cases of leprosy reported, the percentage of multibacillary (MB) cases was low in most of the years investigated, ranging from 9.4% in the year 2000 to 0% in the last five years of the study period (Table 2).

### 3.2. Epidemiological Analysis

Table 2 and Figure 3 and Figure 4 show the total number of cases and incidence rates during the 23 years (2000–2023). The yearly incidence rate ranged from 8.49/1,000,000 population in 2000 to 0.68/1,000,000 population in 2023, remaining within the low-endemic parameter at the end of the study period (<1.00/1,000,000 population), with an overall new case detection rate of 4.46/1,000,000 (Figure 3). Overall, there was a strong downward trend during the study period. Transient increases were observed in 2004 (incidence rate of 5.24/1,000,000), followed by a reduction in cases in 2006 (6.86/1,000,000) and higher peaks in 2007 (7.58/1,000,000) and 2010 (9.13/1,000,000). After a period of decline, with minor peaks in 2015 (2.65/1,000,000), 2019 (2.21/1,000,000), and 2021 (1.25/1,000,000), the incidence rate of leprosy cases in Ecuador showed a downward trend, reaching a minimum value of 0.68/1,000,000 in 2023 (Figure 3).

### 3.3. Spatiotemporal Analysis

Historically, newly detected leprosy cases have been reported in cantons from four provinces in Ecuador: Guayas, Los Rios, Azuay, and El Oro [25]. In total, 36 cantons reported newly detected leprosy cases in the year 2000 (Figure 4), with an annual incidence rate of 8.49 cases per 1,000,000 population. As the year 2000 was the starting baseline for the entire study period, reported one of the highest incidence rates per 1,000,000, and presented a positive spatial autocorrelation value (Moran’s index: 0.03, z-score: 2.83, and *p*-value 0.005), a cluster and outlier analysis (Anselin’s Local Moran’s I statistic) and a hot spot analysis were conducted for this specific year (Figure 5). Negative spatial autocorrelation values were obtained for the remaining study periods. The spatial analysis using a local Moran’s index recognized areas of spatial autocorrelation in the coast region, with High–High clusters in the Guayas and Los Rios provinces and Low–High outliers in the cantons nearby. The spatial association using Getis-Ord-Gi* identified High-Risk clusters for leprosy detection on the coast of Ecuador (Figure 5). Therefore, a more intense clustering/higher concentration of cases (hot spot) was identified on the coast of Ecuador for the year 2000. In the year 2000, one canton from the Loja province presented a high endemicity level (Paltas, NCDR = 464.11), three cantons from the Guayas province reported medium endemicity levels (General Antonio Elizalde, NCDR = 111.95; Balao, NCDR = 110.57; Urbina Jado, NCDR = 102.56), and the remaining cantons presented low endemicity levels, with NCDRs lower than 100 cases per 1,000,000 population (Figure 6). There was a reduction in the number of cases and NCDRs in 2004, especially in the Amazonian region. In 2004, only two cantons reported high endemicity levels (Paute, Azuay province: NCDR = 307.8; Paltas, the Loja province: NCDR = 216.67), while the remaining cantons presented low endemicity levels, with NCDRs lower than 100 cases per 1,000,000 population. There was a resurgence of cases in the Amazonian cantons in 2009, with incidence cases in the canton of the Orellana, Orellana province (NCDR = 94.33), the canton of the Morona, Morona Santiago province (NCDR = 22.91), and the canton of the Shushufindi, Sucumbios province (NCDR = 21.21). However, all cantons presented low endemicity levels this year. In 2014, there was a decline in the number of cases and NCDRs in the coastal cantons and a persistence of cases in the Sierra and Amazonian cantons, such as the Loreto, Orellana province (NCDR = 23.25), the Baba, Los Rios province (NCDR = 22.6), the Rumiñahui, Pichincha province (NCDR = 10.42), the Santo Domingo de los Colorados, Santo Domingo de las Tsáchilas province (NCDR = 9.67), the Loja, Loja province (NCDR = 8.59), the Babahoyo, Los Rios province (NCDR = 5.85), the Quito, Pichincha province (NCDR = 4.77), and the Cuenca, Azuay province (NCDR = 1.75). In the following study periods (2019 and 2023), there was a progressive reduction in cases and NCDRs in the Sierra and coast, with low endemicity levels. By the end of 2023, only seven cantons actively reported cases: the Pasaje, El Oro province (NCDR = 11.27), the Babahoyo, Los Ríos province (NCDR = 10.44), the Milagro, Guayas province (NCDR—9.77), the Ibarra, Imbabura province (NCDR = 6.74), the Machala, El Oro province (NCDR = 3.22), the Santo Domingo de los Colorados, Santo Domingo de los Tsachilas province (NCDR = 2.84), and the Quito, Pichincha province (NCDR = 1.06) (Figure 6). The cantons with the highest number of newly detected leprosy cases during the study period were Guayaquil (364), Quito (195), Babahoyo (109), Daule (63), Milagro (54), and Loja (45), all located in the tropical coast region (Appendix A). Figure 6 shows the decline in the incidence rates reported in the cantons. 

The provinces with the highest number of cases of leprosy during the study period were Guayas (44.8%), Los Rios (15.7%), Pichincha (13.1%), El Oro (4.7%), and Loja (4.4%) (Figure 7). The decline in incidence rates per province and their three respective regions (Costa, Sierra, and Amazonian) are presented in Figure 8 and Figure 9, respectively. Zero cases of leprosy were reported in the insular region (Galapagos Islands) during the study period. The three continental regions of the country presented cases in the year 2000 (with homogeneous regional incidence rates of 1–5/1,000,000), followed by a decline in the number of cases and NCDRs in the Amazon for the year 2004, with a simultaneous increase occurring in the coast (Regional IR = 5–10/1,000,000) (Figure 9). There was a resurgence of cases in the Amazon in 2009, with a persistence of cases occurring in the Andes (Regional IR = 1–5/1,000,000) and coast (Regional IR = 5–10/1,000,000). In 2014, there was a decline in the number of cases and NCDRs on the coast and a persistence of cases in the other two regions (Andes and Amazon—regional IR = 1–5/1,000,000). Interestingly, in the following study periods (2019 and 2023), there was a progressive reduction in cases and NCDRs in the Sierra and coast until the entire country presented regional IR = 0–1/1,000,000. The number of cases per canton, province, and region can be found in the Appendix A.

## 4. Discussion

Our study is unique in that it documents a retrospective official dataset on incidence over a two-decade timespan from a South American country that has effectively applied global guidelines for controlling and eliminating leprosy. Furthermore, we used the spatiotemporal analytic method to study spatial patterns of the disease over a twenty-three-year period at the province and canton levels. Out of the 1539 leprosy cases that were diagnosed and geocoded in Ecuador, male patients were predominant (71.5%) during the study period, with a male-to-female ratio of 2.5. This finding corroborates the literature in Ecuador and worldwide, showing that men are disproportionately affected by this infectious disease [25,26]. One of the first studies ever conducted on leprosy in Ecuador was carried out in 1962 by Blum-Gutierrez [25]. In this study, the author described a predominance of leprosy in males over 15 years of age, especially in the four provinces of Guayas, Los Rios, Azuay, and El Oro [25]. Similarly, our study also found a higher burden of disease in patients over 15 years of age, especially those subjects over 40 years, which accounted for 79% of all the cases (*p* = <0.001). However, it is important to consider that underdiagnosis of leprosy in children under 15 years of age is common in developing countries due to the wide variety of clinical presentations and difficulty in performing in situ clinical peripheral nerve evaluations [27]. Therefore, it is imperative to strengthen clinical diagnosis capabilities to diagnose all cases of leprosy, including atypical cases in pediatric populations. In our study, Guayas also registered the highest number of leprosy cases (44.8%). Regional reports have indicated that leprosy mainly affects the tropics of Ecuador’s Pacific coast, where these endemic provinces are located [28]. 

During the entire study period, the provinces with the highest number of leprosy cases were Guayas (44.8%), Los Rios (15.7%), Pichincha (13.1%), El Oro (4.7%), and Loja (4.4%); three of these provinces are located in the Costa region, and the remaining two are located in the Sierra region. Historically, leprosy cases in Ecuador have been concentrated in provinces and rural cantons from the Costa region. Ecuador’s significant urbanization began in the first half of the 20th century [29]. This trend continues, with large cities, including Guayaquil and Quito, absorbing a large portion of the rural population over the last few decades [30]. Rural populations from agricultural regions and peri-urban settlements migrate to larger cities, such as Quito and Guayaquil, seeking employment, education, better access to healthcare, and living conditions [30]. People moving from rural to urban areas may bring previously undiagnosed cases into cities, where they are more likely to be detected due to better medical care. This can contribute to fluctuations in case numbers, as rural regions may have lower detection capacities. Moreover, in areas of Ecuador with higher numbers of immigrants from neighboring countries, leprosy cases may rise if migrants come from Andean regions with a higher endemicity status (e.g., Venezuela, Colombia, and Bolivia) [31]. Ecuador’s borders with Colombia and Peru, for instance, could be focal points for this type of migration-related fluctuation. Moreover, in rural areas of the Pacific coast and the Amazonian rainforest of Ecuador, the hunting, handling, and consumption of armadillos are common practices. Armadillos are known to be a natural reservoir of *Mycobacterium leprae*, and the connection between armadillos and leprosy is particularly noted with *Dasypus novemcinctus* [32]. *Dasypus novemcinctus* is mainly found in the rural coastal regions of Ecuador, which include tropical and dry forests, such as those in the Manabí and Guayas provinces. The Amazon rainforest is also home to armadillos, but the human population density is very low [33]. Leprosy remains predominantly associated with tropical regions and is considered by the WHO as a neglected tropical disease (NTD).

Our results are further corroborated by past systematic reviews and meta-analyses that concluded that males tend to present a higher prevalence of leprosy in high-burden countries [34]. There is also evidence that increasing age is associated with a greater risk of leprosy [34]. Several hypotheses indicate gender and age differences regarding the severity and disability produced by *M. leprae* infections. Brazilian studies have suggested that the higher burden of leprosy in the elderly can be explained by the higher life expectancy achieved, which might result in a higher number of new cases in this age group [35]. Perhaps men are infected before women and manifest the signs and symptoms earlier [3]. The time between the appearance of the clinical manifestations of leprosy and the diagnosis can vary from many months to several years in developing countries [3]. Therefore, late diagnosis can be another explanation for the over-representation of elderly patients in Ecuador. The appearance of MB cases in Brazil was directly proportional to increased age due to the long incubation period of leprosy combined with a late diagnosis [36]. Additionally, relapses of MB cases are more common in older age groups [37,38] and males, probably because of the higher prevalence observed in this gender [38]. Other studies have suggested more social contact between the elderly populations in South American countries, which might increase the predominance of leprosy among elderly age groups [35,38]. As immune senescence occurs with aging, more severe clinical presentations are observed in the elderly population which sometimes go un-noticed [39] and can act as a reservoir of infection in vulnerable populations [40]. 

The diagnosis of leprosy is based on skin lesions, nerve thickness, and microscopic detection of bacilli and is classified into two types: paucibacillary (PB) cases and multibacillary (MB) case [7]. However, in the present study, the cases were classified according to Ridley and Joplin (1966) and the ICD-10-CM Code for Leprosy [Hansen’s disease] A30. Cases of indeterminate leprosy, tuberculoid leprosy (TT), and borderline tuberculoid leprosy (BT) were grouped in the paucibacillary (PB) group, while borderline borderline (BB), borderline lepromatous (BL), and lepromatous (LL) cases were classified under the multibacillary (MB) group [7]. Unfortunately, 73% of the reported cases were unspecified in this study. This can be explained by the changes in the reporting systems of leprosy throughout the study period. For instance, the histological classification of leprosy was most widely used during the first two years of the study period. Posteriorly, the Ecuadorian health authorities adopted the recommended clinical classification by PAHO/WHO (Paucibacilalry/Multibacillary) [6]. Nationally, both classifications are used and recognized. However, the microscopic classification is more efficiently implemented in urban areas with access to clinical laboratory diagnosis, while the clinical classification by PAHO/WHO is more practical in rural areas with poor access to laboratory services. Regardless of the clinical classification used by the Ecuadorian Ministry of Public Health (MSP), it is important to recognize that Ecuador is a developing country, with 36% of its population living in remote rural areas [41]. In these scenarios, the availability of microscopy is limited; therefore, the microscopic differentiation in the tuberculoid–lepromatous spectrum is impossible. Additionally, technological advances also played a role in the reporting strategies in Ecuador. From the year 2000–2012, obligatory notifiable diseases (such as leprosy) were reported manually through the submission of the form EPI-2 by healthcare workers. In 2013, the MSP implemented the online “Integrated System of Epidemiological Surveillance” (Sistema Integrado de Vigilancia Epidemiológica—SIVE) and a health emergency notification system called SIVE-ALERTA [42]. Modernizing the reporting system improved the efforts to efficiently detect and control infectious diseases such as leprosy. During the study period, only 120 cases were classified as MB (116 LL cases, 4 BL cases, and 12 BB cases). Interestingly, 104 (86.7%) of the cases were detected in the Andean region, with most of the cases coming from the Pichincha province. Additionally, the Santo Domingo de los Tsachilas province (located on the coast) was administratively part of Pichincha province before November 2007. Therefore, before 2007, leprosy cases from Santo Domingo were added to the Pichincha province. The findings suggest a possible cluster of MB cases or an increased capability of clinical/microbiological diagnosis in the coast region. 

The year 2000 was a remarkable year for the history of leprosy control globally, as the elimination of leprosy as a public health problem was achieved this year (defined as a world prevalence of less than 1 per 10,000 population), as declared by the World Health Assembly resolution 44.9 and in most countries by 2010 [43]. However, Ecuador was classified as a country with a high burden of leprosy in the Americas for the years 2010–2011 (reporting 100 or more new cases per year) [44]. Favorably, Ecuador has shown positive action strategies for reaching regional goals, as indicated by its most recent leprosy indicators. Ecuador follows the regional WHO recommendations, such as the Pan American Health Organization (PAHO) report (CD49.R19) “Elimination of Neglected Diseases and Other Poverty-Related Infections” [19]. The main goal of this report is to eliminate leprosy as a public health problem (<1 case per 10,000 people) from the first sub-national political/administrative level [45]. Additionally, Ecuador has been following the regional guidelines to prevent and control leprosy transmission through active surveillance/monitoring and administration of multidrug therapy (MDT) to cure affected patients and reduce transmission in the communities [46]. Leprosy patients are examined every six months for a five-year period, and family members or other exposed individuals are also followed over time to search for clinical or laboratory manifestations of the disease [19]. Additionally, the patient, family members, and other exposed individuals are actively being educated on the forms of disease transmission, the incubation period, the signs and symptoms of leprosy, and the actions to take in case of presenting the clinical manifestations [19]. The MSP leprosy management guidelines are aligned with the WHO multidrug therapy (MDT) combination treatment composed of dapsone, rifampicin, and clofazimine, with a duration of 6 months being required for PB leprosy and 12 months for MB leprosy [6]. Hospitalization is indicated only in cases of severe adverse reactions to the MDT treatment in Ecuador [19]. Interestingly, the BCG vaccine also offers partial protection against non-tuberculosis mycobacterial infections such as leprosy [47]. The protective properties of the BCG vaccine against *M. leprae* infection involves the cross-reactivity of B cells and T cells against mycobacterial antigens shared between mycobacterial species [48]. The WHO and the Strategic Advisory Group of Experts (SAGE) agree that BCG at birth is effective at reducing the risk of *M. leprae* infection and therefore should be maintained as a preventive strategy in high-burden countries [6]. The national leprosy guidelines also recommend maintaining high coverage of the Bacillus Calmette–Guérin (BCG) vaccination nationwide as another measure to reduce the burden of leprosy in Ecuador [19]. Recent reports have indicated that the BCG vaccination coverage was 86.5% during the 2010–2020 period [49]. We believe that these control strategies implemented in the previous decades (BCG vaccination, early detection of leprosy cases, treatment with MDT combination, public health education, and follow-up of infected patients and close relatives) have greatly contributed to the reduced cases reported in Ecuador. Moreover, other strategies can be strengthened to secure the elimination of leprosy nationwide, such as improving access or referral to essential care services, controlling zoonotic transmission of *M. leprae* (nine-banded armadillo *Dasypus novemcinctus*), reinforcing the health information systems nationwide/reducing stigma and/or discrimination against persons affected by the disease, and reducing the delay in case detection in rural areas [50]. This study presented several limitations. First, population data were calculated based on national and province growth rates reported by the Ecuadorian Agency of Statistics (INEC). Therefore, we did not consider possible fluctuations caused by urbanization and other forms of migration. However, the Ecuadorian population is known to have been growing at a relatively stable rate in the past twenty years [51]. Second, potential risk factors such as low socioeconomic status [52,53], poor living conditions [12,54], low access to healthcare [54], low household income [55], race/ethnicity [56], and the presence of co-infections [56,57], which have been previously reported as being associated with a high prevalence of *M. leprae* infection, were not incorporated in this study. Third, the retrospective nature of this study did not allow us to verify the data in real-time. Therefore, the case notification rate might not be representative of the incidence and the infection trends might be over- or underestimated.

## 5. Conclusions

We conclude that Ecuador has effectively applied global guidelines for the control and elimination of leprosy in the past twenty-three years, as a strong downward trend was observed during the study period. However, a few cases remain in some cantons from the Pacific coast and the Sierra region of the country. Therefore, it is imperative to maintain surveillance, prevention, and control strategies in those cantons in regard to higher risk of leprosy detection and protecting the most vulnerable populations, such as male adults over 40 years from the Costa and Sierra regions (such as cantons from the El Oro, Santo Domingo de los Tsachilas, Los Rios and Imbabura provinces). While tangible progress is being made, continued efforts are needed to halt the transmission of leprosy in Ecuador and ultimately reach the goal of leprosy elimination.

## Figures and Tables

**Figure 1 tropicalmed-09-00246-f001:**
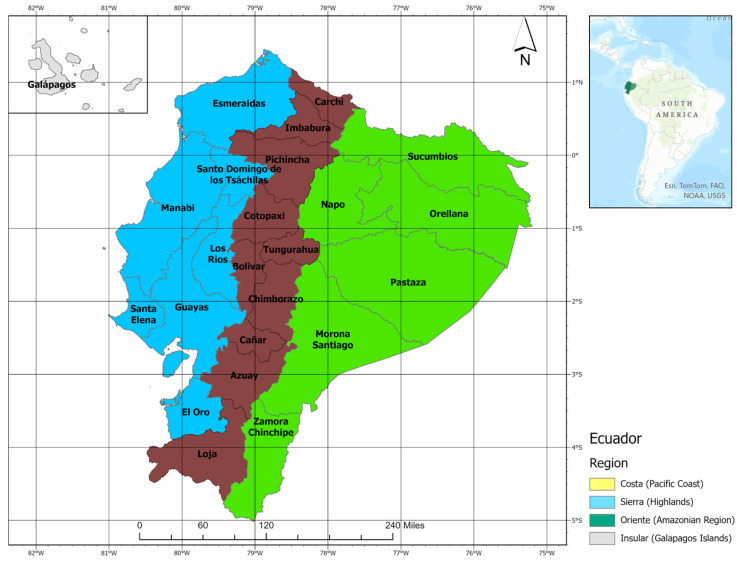
Administrative Regions and Provinces of Ecuador. Continental Ecuador is bisected from north to south by the Andes highlands. The country has four ecoregions (the coast, Andes, Amazon, and the Galapagos Islands). In the Pacific coast region, there are seven provinces (colored in yellow). In the Andes region, there are 10 provinces (colored in blue), and in the Amazon basin region, there are six provinces (colored in green). Source: Adapted from Instituto Nacional de Estadísticas y Censos (https://www.ecuadorencifras.gob.ec/institucional/home/ (accessed on 12 January 2024)).

**Figure 2 tropicalmed-09-00246-f002:**
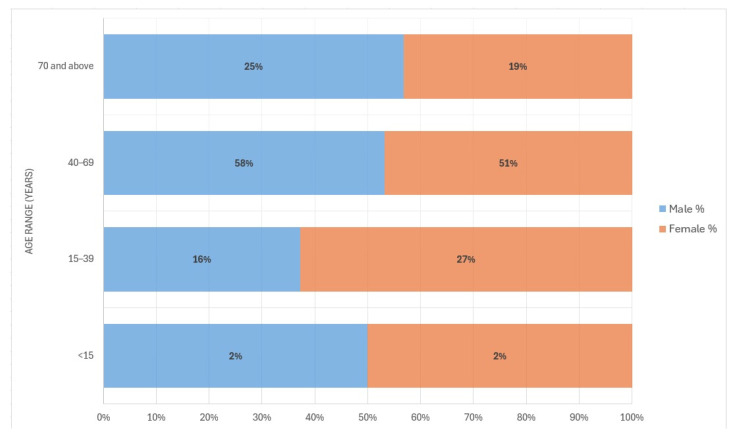
Age and gender of detected leprosy cases in Ecuador (2000–2023). Source: Ministerio de Salud Pública (MSP). Percentage of age range by gender.

**Figure 3 tropicalmed-09-00246-f003:**
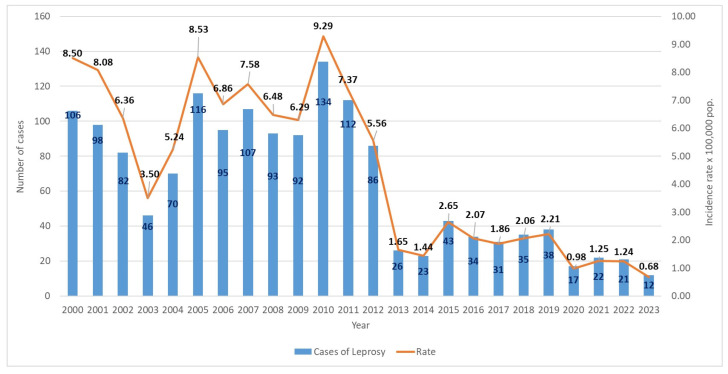
Annual number of cases and incidence rates (/1,000,000) of leprosy in Ecuador from 2000 to 2023. The incidence rates were placed above the bars and the number of cases was placed inside the bars of the figure. Source: Ministerio de Salud Pública (MSP).

**Figure 4 tropicalmed-09-00246-f004:**
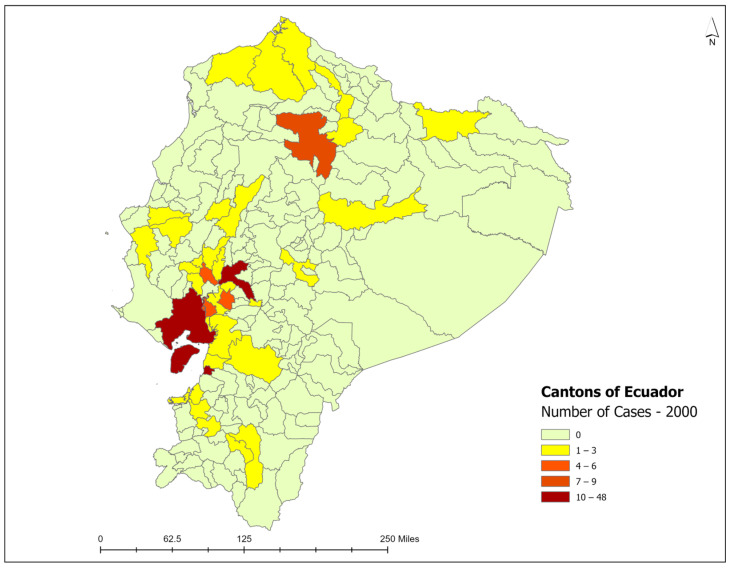
Number of cases of leprosy by cantons in Ecuador for the year 2000. Source: Ministerio de Salud Pública (MSP).

**Figure 5 tropicalmed-09-00246-f005:**
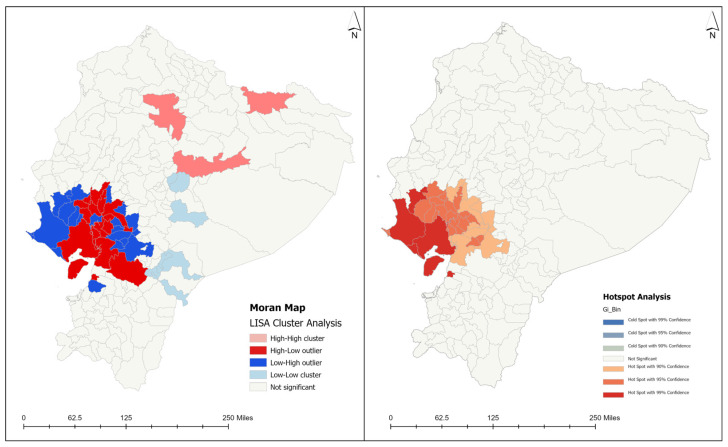
Cluster and outlier analysis and hot spot analysis for the number of leprosy cases reported in the year 2000. Source: Ministerio de Salud Pública (MSP).

**Figure 6 tropicalmed-09-00246-f006:**
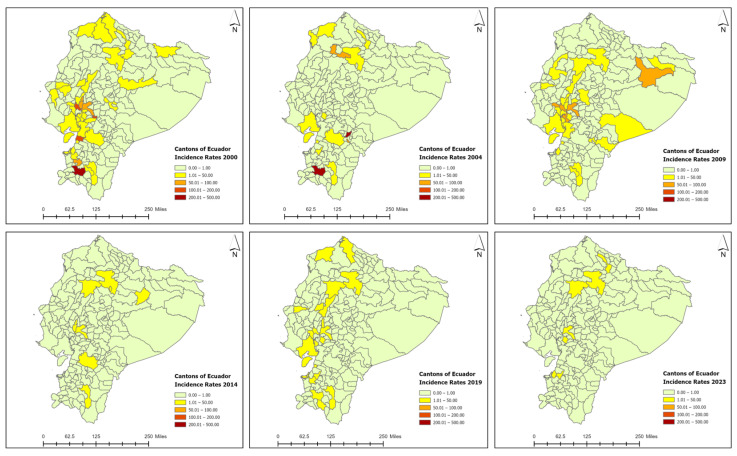
Incidence rates of leprosy by cantons in Ecuador, 2000–2023. Source: Ministerio de Salud Pública (MSP).

**Figure 7 tropicalmed-09-00246-f007:**
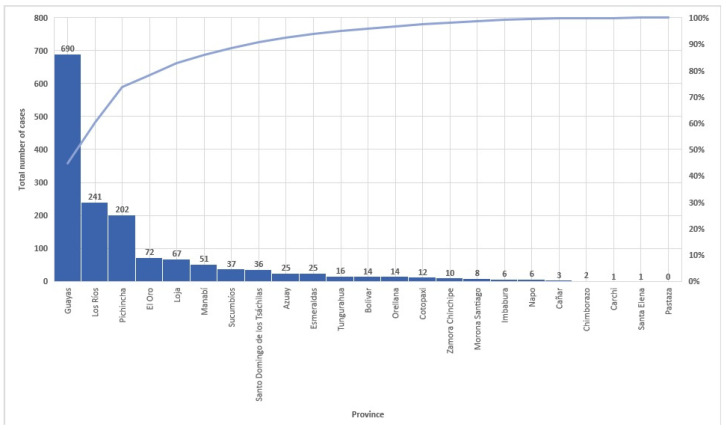
Total number of leprosy cases in Ecuador by provinces (2000–2023). Source: Ministerio de Salud Pública (MSP).

**Figure 8 tropicalmed-09-00246-f008:**
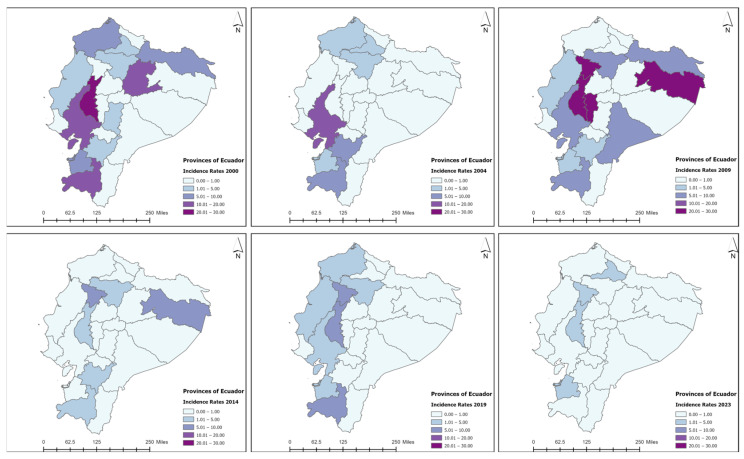
Incidence rates of leprosy by province in Ecuador, 2000–2023. Source: Ministerio de Salud Pública (MSP).

**Figure 9 tropicalmed-09-00246-f009:**
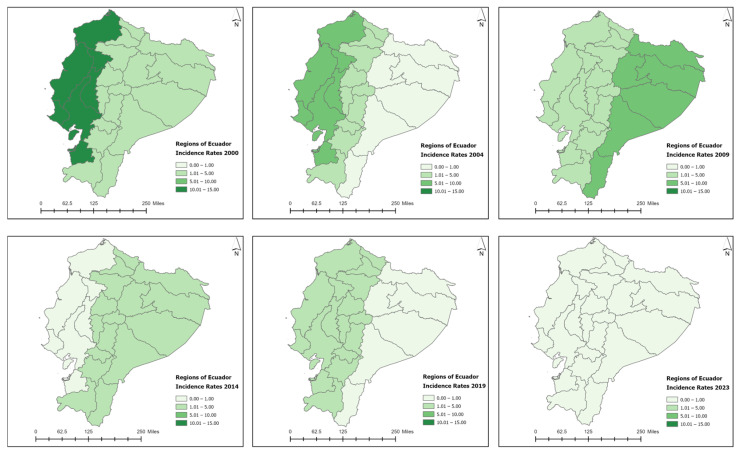
Incidence rates of leprosy by region in Ecuador, 2000–2023. Source: Ministerio de Salud Pública (MSP).

**Table 1 tropicalmed-09-00246-t001:** Frequency of sociodemographic and clinical classifications of leprosy patients evaluated in the study.

Variable	Coast (N = 1070)	Sierra (N = 446)	Amazon (N = 23)	Total (N = 1539)	*p*-Value
Age *	55 (18)	53 (17)	39 (25)	54 (18)	<0.001
Age group					<0.001
<15	21 (2.0%)	1 (0.2%)	8 (34.8%)	30 (1.9%)	
15–39	185 (17%)	107 (24%)	2 (8.7%)	294 (19.1%)	
40–69	600 (56%)	250 (56.1%)	11 (47.8%)	861 (56.0%)	
70 and above	264 (25%)	88 (19.7%)	2 (8.7%)	354 (23.0%)	
Gender					0.379
Female	293 (27%)	139 (31%)	7 (30%)	439 (28.5%)	
Male	777 (73%)	307 (69%)	16 (70%)	1100 (71.5%)	
Status					0.585
Ambulatory (Outpatient)	393 (37%)	155 (35%)	10 (43%)	558 (36%)	
Hospitalized	677 (63%)	291 (65%)	13 (57%)	981 (64%)	
ICD Classification					<0.001
Borderline	4 (0.4%)	8 (1.8%)	0 (0%)	12 (0.8%)	
Borderline Lepromatous	1 (0.1%)	3 (0.7%)	0 (0%)	4 (0.3%)	
Borderline Tuberculoid	8 (0.7%)	10 (2.2%)	0 (0%)	18 (1.2%)	
Indeterminate	175 (16%)	69 (15%)	5 (22%)	249 (16.2%)	
Lepromatous	15 (1.4%)	101 (23%)	0 (0%)	116 (7.5%)	
Other	3 (0.3%)	0 (0%)	0 (0%)	3 (0.2%)	
Tuberculoid	8 (0.7%)	6 (1.3%)	0 (0%)	14 (0.9%)	
Unspecified	856 (80%)	249 (56%)	18 (78%)	1123 (73%)	
WHO Classification					<0.001
Paucibacillary (PB)	1050 (98.1%)	334 (74.9%)	23 (100%)	1407 (92%)	
Multibacillary (MB)	20 (1.9%)	112 (25.1%)	0 (0%)	132 (8%)	

Source: Ministerio de Salud Pública (MSP). * Mean (sd)—one-way ANOVA performed for continuous variable; Chi-square or Fisher–Freeman–Halton exact test for categorical variables; Fisher–Freeman–Halton exact test performed for variables with observations with counts < 5. *n* (%).

**Table 2 tropicalmed-09-00246-t002:** Trends in leprosy cases in Ecuador from 2000 to 2023.

Year	New Cases Detected	NCDR per 1,000,000 Population. *	MB Proportion. n (%). *		Children under 15 Proportion. n (%)		Female Proportion. n (%)	
2000	106	8.50	10	9.4	0	0%	35	33%
2001	98	8.08	14	14.3	1	1.0%	32	33%
2002	82	6.36	12	14.6	0	0%	30	37%
2003	46	3.50	5	10.9	1	2.2%	16	35%
2004	70	5.24	16	22.9	2	2.9%	18	26%
2005	116	8.53	9	7.8	6	5.2%	44	38%
2006	95	6.86	4	4.2	1	1.1%	36	38%
2007	107	7.58	5	4.7	6	5.6%	36	34%
2008	93	6.48	4	4.3	0	0%	22	24%
2009	92	6.29	11	12.0	0	0%	22	24%
2010	134	9.29	12	9.0	7	5.2%	16	12%
2011	112	7.37	5	4.5	5	4.5%	16	14%
2012	86	5.56	17	19.8	0	0%	33	38%
2013	26	1.65	3	11.5	0	0%	9	35%
2014	23	1.44	2	8.7	0	0%	11	48%
2015	43	2.65	2	4.7	0	0%	16	37%
2016	34	2.07	0	0.0	0	0%	3	8.8%
2017	31	1.86	0	0.0	0	0%	9	29%
2018	35	2.06	1	2.9	0	0%	8	23%
2019	38	2.21	0	0.0	1	2.6%	9	24%
2020	17	0.98	0	0.0	0	0%	4	24%
2021	22	1.25	0	0.0	0	0%	9	41%
2022	21	1.24	0	0.0	0	0%	4	19%
2023	12	0.68	0	0.0	0	0%	1	8.3

Source: Ministerio de Salud Pública (MSP). * NCDR: New Case Detection Rate. MB: Multibacillary.

## Data Availability

Data were obtained from publicly available databases from the Ecuadorian National Institute of Statistics and Census (INEC) and the Ecuadorian Ministry of Public Health (MSP). By legal mandate, all records are de-identified. A summary of the data is provided in Appendix A.

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
