# Peer review of "Ecuador Towards Zero Leprosy: A Twenty-Three-Year Retrospective Epidemiologic and Spatiotemporal Analysis of Leprosy in Ecuador"

_tropicalmed, 2024, doi:10.3390/tropicalmed9100246_

Round 1

Reviewer 1 Report

Comments and Suggestions for Authors

The article titled "Ecuador Towards Zero Leprosy: A Twenty-Three-Year Retrospective Epidemiologic and Spatiotemporal Analysis of Leprosy in Ecuador" presents an in-depth examination of the trends, incidence, and spatial distribution of leprosy in Ecuador from 2000 to 2023. The study employs a robust methodological approach, combining epidemiologic and spatiotemporal analysis to offer a comprehensive view of how leprosy cases have evolved in Ecuador. The manuscript is well-written and easy to follow. Nevertheless, some issues should be addressed. In particular:

1. Use passive voice.

2. Use the right format to cite references (References must be numbered in order of appearance in the text (including table captions and figure legends) and listed individually at the end of the manuscript. We recommend preparing the references with a bibliography software package, such as EndNote, ReferenceManager or Zotero to avoid typing mistakes and duplicated references. We encourage citations to data, computer code and other citable research material.).- Use brackets!

3. The captions for the figures should be after the figures.

4. Table 1 has not have the right format as Table 2 for istance.

5. In section 2.2 (Study Subjects), provide a brief explanation for some specific terms such as "paucibacillary" (PB) and "multibacillary" (MB) infections for readers unfamiliar with these classifications.

6. In Figure 1 and other figures, enhance the color contrast to improve the readability of the ecoregions and provinces. It might be difficult for some readers to distinguish between regions on the map, especially in print versions.

7. In section 2.5, further emphasize that the data used is anonymized and publicly available.

8. The discussion could benefit from tying the findings more explicitly back to global leprosy elimination efforts.

9. When discussing the Hot Spot Analysis and Moran’s Index (Section 3.3), a brief summary of the significance of these results in layman's terms would help general readers.

10. Ensure consistent use of terms such as "leprosy burden" and "leprosy incidence" throughout the text.

11.  In Section 3.1, where you mention the differences between age groups, providing a more detailed explanation as to why leprosy incidence is significantly higher in the older age groups could strengthen the discussion.

12. Currently, the manuscript does not include a clear conclusions section. I suggest adding a separate conclusions section that briefly summarizes the key findings of the study and their implications for leprosy control and public health policy. This section could also highlight future research directions or recommendations for addressing the remaining challenges in leprosy elimination in Ecuador and beyond. A well-defined conclusion will provide a more complete and cohesive ending to the manuscript.

Comments on the Quality of English Language

A minor editing of the English language is required.

Author Response

Comments 1: Use Passive voice

Response 1: Thank you for pointing this out. We agree with this comment. Therefore, we have changed the text using passive voice (“in quotes”) and improved the grammar in the following row numbers:

1.       Introduction:

-          Leprosy is “classified “as a neglected tropical disease (NTD) “and is reported to have”.. [Row # 37]

-          Leprosy, also known as Hansen’s disease, is “characterized as” a.. [Row # 41]

-          “by the” slow-growing acid-fast “bacilli”, Mycobacterium “leprae” [Row # 42]

-          “Transmission” occurs through respiratory droplets “released by” coughing or sneezing during prolonged close contact [Row # 43]

-          The microorganism is slow growing, “with” the incubation period “ranging from” 2 to 12 years, “and” an estimated average of five years. “Variations in the incubation period depend” on the clinical form of the disease, “appearing” shorter for paucibacillary (PB) disease (2-5 years) “compared to” multibacillary (MB) disease [Row # 46-50]

-          “The” ICD-10-CM Code for Leprosy [Hansen's disease] is [Row # 53]

-          PB leprosy cases “are characterized by” 1-5 skin lesions, without the demonstrated presence of bacilli in the skin smear. “In contrast”, MB cases “are identified with” more than 5 skin lesions; nerve involvement (pure neuritis, or any number of skin lesions and neuritis); or “the” demonstrated presence of bacilli in a slit-skin smear, irrespective of the number of lesions. [Row # 56-60]

-          BT cases can be included in the PB group, “while” BB, BL, and LL patients can be classified under the MB group. This chronic infectious disease is curable, “and disability can be prevented” if early treatment is provided. [Row # 61-63]

-          The WHO guidelines for the diagnosis, treatment, and prevention of leprosy are followed in Ecuador. [Row # 65-66]

-          Since the recommendation of MDT in 1981 by the WHO, “the” worldwide [Row # 68]

-          However, despite the decrease in case notification rates, “it has been suggested by” epidemiological and spatiotemporal studies done in Brazil and China “that” leprosy has focal distribution with potential for hidden cases, “necessitating” intensified targeted control activities to reach the goal of “the” ‘Towards Zero Leprosy’ strategy [Row # 73-77]

-          In a study conducted in the Philippines in 2013, static, low transmission of leprosy in endemic rural areas and among children was identified despite good MDT coverage, and a new approach of chemoprophylaxis/immunoprophylaxis was suggested to target leprosy elimination [Row # 77-80]

-          demographic factors such as “being male” and “older age” are the [Row # 81-82]

-          In 2020, the highest proportion of cases was detected in the Southeast Asian region, followed by the Americas, with “74% of all worldwide new cases reported by” Brazil, India, and Indonesia. A new case detection rate of 18.8 per million population was recorded in the Americas, which is higher than the worldwide new case detection rate of 16.4 per million population. [Row # 83-87]

-          “Still, in the last 10 years,” there have been [Row # 92-93]

-          A leprosy notification system called SIVE-ALERTA is used by the Ecuadorian Ministry of Public Health (MSP) through an electronic PAHO-supported platform called ViEpi (Vigilancia Epidemiológica) [Row # 99-101]

-          Currently, new cases are reported through the SIVE-ALERTA system using both passive and active notification strategies [Row # 103-104]

-          Focal areas with transmission and vulnerable populations will also be identified, which can be specifically targeted to achieve the goal of zero autochthonous cases in line with WHO's “Towards Zero Leprosy’ strategy” [Row # 107-109]

2.       Materials and Methods

-          Distinct ecological diversity is found in each region [Row # 117].

-          In recent years, the MSP passive and active notification strategies “were used” to report new cases through the SIVE-ALERTA system 35. [Row # 148-149]

-          PB leprosy cases “were characterized by” 1-5 skin lesions, without the demonstrated presence of bacilli in the skin smear. MB cases “were characterized by over 5 skin lesions; nerve involvement; or the demonstrated presence of bacilli in a slit-skin smear, irrespective of the number of lesions [Row # 163-168]

-          Descriptive statistics “such as” frequency (%) for categorical variables and median [Row # 190]

3.       Results

-          “One hundred” and four of the cases (86.7%) were detected in the Sierra region [Row # 268-269]

-          The provinces with the highest number of cases of leprosy during the study period were Guayas (44.8%), Los Rios (15.7%), Pichincha (13.1%), El Oro (4.7%), and Loja (4.4%) [Row # 337-338]

4.       Discussion

-          Brazilian studies have “suggested” that the higher burden … [Row # 436]

-          The appearance of MB cases in Brazil was “directly proportional” to increased age, due to the long incubation period of leprosy, combined with a late diagnosis [Row # 442-443]

-          As immune-senescence occurs with aging, more severe clinical presentations are observed in the elderly population, which “sometimes go” unnoticed .. [Row # 448-449]

-          Cases of indeterminate leprosy, tuberculoid leprosy (TT), and borderline-tuberculoid leprosy (BT) were grouped in the paucibacillary (PB) group, “while” borderline-borderline (BB), borderline-lepromatous (BL), and lepromatous (LL) cases were classified under the multibacillary (MB) group [Row # 455-458]

-          Additionally, Santo Domingo de los Tsachilas province (located “on the” Coast) [Row # 481-482]

-          “Therefore”, the case notification rate might not be representative “of” the incidence, and the infection trends might be over or underestimated [Row # 540-541]

Comments 2: Use the right format to cite references (References must be numbered in order of appearance in the text (including table captions and figure legends) and listed individually at the end of the manuscript. We recommend preparing the references with a bibliography software package, such as EndNote, ReferenceManager, or Zotero to avoid typing mistakes and duplicated references. We encourage citations to data, computer code, and other citable research material.). - Use brackets!

Response 2: Agree. We have revised the in-text citations and references and formatted the document according to the journal’s guidelines: use of square brackets for the in-text citation, numbered in order of appearance. References were numbered in order of appearance in the text. EndNote was used as a Reference manager for the manuscript. Observation: Even though all references appear in order of appearance, some references are used again later in the text. For example, if the second reference is mentioned after the first 8 references, [1-8], then reference #2 is cited even after the eighth one [2]. However, we followed the chronological order and never cited in reverse order.

Comments 3: The captions for the figures should be after the figures.

Response 3: We agree with your observation and have accordingly placed the Figure title and caption after the figures. We have done so for Figure 1 (Page 5), Figure 2 (Page 8), Figure 3 (Page 10), Figure 4 (Page 12), Figure 5 (Page 12), figure 6 (Page 13), Figure 7 (Page 13), Figure 8 (Page 14), and Figure 9 (Page 14).

Comments 4: Table 1 has not have the right format as Table 2 for instance.

Response 4: Thank you for pointing this out. We agree with this comment. Therefore, we have formatted Table 1 according to the journal’s guidelines. The changes can be found on Page 7 (Results section), [Row #255].

Comments 5: In section 2.2 (Study Subjects), provide a brief explanation for some specific terms such as "paucibacillary" (PB) and "multibacillary" (MB) infections for readers unfamiliar with these classifications.

Response 5: Thank you for pointing this out. We described the clinical leprosy classification (Paucibacillary and Multibacillary) under the “Study subjects” section. We added the following explanation: The paucibacillary (PB) or multibacillary (MB) classification was based on the number of skin lesions, presence of nerve involvement, and identification of bacilli on slit-skin smear. PB leprosy cases presented 1-5 skin lesions, without the demonstrated presence of bacilli in the skin smear. MB cases presented over 5 skin lesions; or nerve involvement; or with the demonstrated presence of bacilli in a slit-skin smear, irrespective of the number of lesions (Row #163-168) – “2. Materials and methods” – “2.2. Study subjects”. Additionally, the clinical classification was described in the background section: [Row # 53 – 66].

Comments 6: In Figure 1 and other figures, enhance the color contrast to improve the readability of the ecoregions and provinces. It might be difficult for some readers to distinguish between regions on the map, especially in print versions.

Response 6: Thank you for this important observation. We changed the colors of the three ecoregions in Figure 1 from east to west: “Big Sky Blue”, “Leather Brown”, and “Quetzal Green”, to represent water on the Pacific Coast, Elevation in the Andes, and Vegetation in the Amazonian region. [Page 5. Figure 1. Row # 171].

Comments 7: In section 2.5, further emphasize that the data used is anonymized and publicly available.

Response 7: Thank you for your observation. We added the following text to section “2.5. Ethics statement”: Therefore, the datasets used in this study did not report personally identifiable information, such as name, national identification number, date/place of birth, or biometric records.

Comments 8: The discussion could benefit from tying the findings more explicitly back to global leprosy elimination efforts.

Response 8: Thank you for your observation. We added text to the “Discussion” section to link our results with global leprosy elimination efforts: “The year 2000 was a remarkable year for the history of leprosy control globally, as the elimination of leprosy as a public health problem was achieved this year (defined as a world prevalence of less than 1 per 10,000 population) as declared by the World Health Assembly resolution 44.9 and in most countries by 2010. However, Ecuador was classified as a country with a high burden of leprosy in The Americas for the year 2010-2011 (reporting 100 or more new cases per year). Favorably, Ecuador has shown positive action strategies for reaching regional goals, as indicated by its most recent leprosy indicators. [Row #487-494]

Comments 9: When discussing the Hot Spot Analysis and Moran’s Index (Section 3.3), a brief summary of the significance of these results in layman's terms would help general readers.

Response 9: We appreciate your observation. We added the following description after explaining the results of the hotspot analysis: “Therefore, a more intense clustering/higher concentration of cases (hot spot) was identified on the coast of Ecuador for the year 2000” [Row #335-336].

Comments 10: Ensure consistent use of terms such as "leprosy burden" and "leprosy incidence" throughout the text.

Response 10: Thank you for this observation. To clarify the difference in both terms, we added an explanation of these differences in section 2.4. Spatial and Spatiotemporal analysis: [“Leprosy endemicity was measured by the new case detection rate (NCDR) and new cases among children. Leprosy burden refers to a broader concept of the impact of the disease in a community and includes NCDR, MB cases, cases in children/women, societal consequences of the persons living with the disability (social stigma), and prevalence”]. [Row #212-216]

Comments 11: In Section 3.1., where you mention the differences between age groups, providing a more detailed explanation as to why leprosy incidence is significantly higher in the older age groups could strengthen the discussion.

Response 11: Thank you for your observation. Since we described our findings in “Section 3.1. Study population”, we discussed these results further in the Discussion section:

-          Similarly, our study also found a higher burden of disease in patients over 15 years of age, especially those subjects over 40 years, which accounted for 79% of all the cases (p=<0.001) [Row #399-401].

-          The time between the appearance of the clinical manifestations of leprosy and the diagnosis can vary from many months to several years in developing countries. Therefore, late diagnosis can be another explanation for the overrepresentation of elderly patients in Ecuador. The appearance of MB cases in Brazil was directly proportional to increased age, due to the long incubation period of leprosy, combined with a late diagnosis. Additionally, relapses of MB cases are more common in older age groups, and males, probably because of the higher prevalence observed in this gender. Other studies have suggested more social contact between the elderly population in South American countries, which might increase the predominance of leprosy among elderly age groups. As immune-senescence occurs with aging, more severe clinical presentations are observed in the elderly population, which sometimes go unnoticed, and can act as a reservoir of infection in vulnerable populations [Row #439-450].

Comments 12: Currently, the manuscript does not include a clear conclusions section. I suggest adding a separate conclusions section that briefly summarizes the key findings of the study and their implications for leprosy control and public health policy. This section could also highlight future research directions or recommendations for addressing the remaining challenges in leprosy elimination in Ecuador and beyond. A well-defined conclusion will provide a more complete and cohesive ending to the manuscript.

Response 12: Thank you for pointing this out. We agree with this comment. Therefore, we have added a “5. Conclusion” section: “We conclude that Ecuador has effectively applied global guidelines for the control and elimination of leprosy in the past twenty-three years, as a strong downward trend was observed during the study period. However, few cases remain in some cantons from the Pacific Coast and the Sierra region of the country. Therefore, it is imperative to maintain surveillance, prevention, and control strategies in those cantons with higher risk of leprosy detection and the most vulnerable populations, such as male adults over 40 years from the Costa and Sierra regions (such as cantons from El Oro, Santo Domingo de los Tsachilas, Los Rios and Imbabura provinces). While tangible progress is being made, continued efforts are needed to halt the transmission of leprosy in Ecuador and ultimately reach the goal of leprosy elimination”. [Row #551-560]

4. Response to Comments on the Quality of English Language

Point 1: A minor editing of the English language is required.

Response 1: The edits were described in detail in the response to Comment 1 (on the first page of this document).

5. Additional clarifications

References were numbered in order of appearance in the text. EndNote was used as a Reference manager for the manuscript. Observation: Even though all references appear in order of appearance, some references are used again later in the text. For example, reference #7 may appear after reference #50, because reference #7 also applies to the explanation of that point in the discussions section.

Reviewer 2 Report

Comments and Suggestions for Authors

Thank you for the opportunity to review this article. The author conducted a study focusing on leprosy in Ecuador. While the topic is of significant social importance, the manuscript has fundamental methodological issues. Therefore, I believe it requires extensive revision before it can be considered further. I have outlined my comments below.

Comments:

1.       The authors should check for typos, especially those marked in bold in the main text.

Methods:

2.       The authors should add the description about the four ecological regions. It is informative for potential reviewers of this manuscript.

3.       The reviewer acknowledges that the authors used official leprosy case data, but the accuracy of this data is crucial. The authors should provide further details regarding its accuracy.

4.       Does the surveillance system adequately cover all leprosy cases?

5.       Were there any changes in the reporting system during the study period? If so, the authors should describe them. This could also be a limitation of the study.

Results:

6.       The authors should adjust Figure 3 to make the numbers easier to read.

Discussion:

7.       The authors should provide a more detailed discussion of the regions/provinces with a high number of leprosy cases.

8.       Why were 73% of the reported cases unspecified in this study?

9.       The authors should provide more detailed descriptions of 'urbanization and other forms of migration' in Ecuador.

10.       The authors should include a concluding statement for this study.

Comments on the Quality of English Language

None.

Author Response

Comments 1: The authors should check for typos, especially those marked in bold in the main text.

Response 1: Thank you for pointing this out. We agree with this comment. Therefore, we have changed the text using passive voice (“in quotes”) and improved the grammar (“typos”) in the following row numbers:

1.       Introduction:

-          Leprosy is “classified “as a neglected tropical disease (NTD) “and is reported to have”.. [Row # 37]

-          Leprosy, also known as Hansen’s disease, is “characterized as” a.. [Row # 41]

-          “by the” slow-growing acid-fast “bacilli”, Mycobacterium “leprae” [Row # 42]

-          “Transmission” occurs through respiratory droplets “released by” coughing or sneezing during prolonged close contact [Row # 43]

-          The microorganism is slow growing, “with” the incubation period “ranging from” 2 to 12 years, “and” an estimated average of five years. “Variations in the incubation period depend” on the clinical form of the disease, “appearing” shorter for paucibacillary (PB) disease (2-5 years) “compared to” multibacillary (MB) disease [Row # 46-50]

-          “The” ICD-10-CM Code for Leprosy [Hansen's disease] is [Row # 53]

-          PB leprosy cases “are characterized by” 1-5 skin lesions, without the demonstrated presence of bacilli in the skin smear. “In contrast”, MB cases “are identified with” more than 5 skin lesions; nerve involvement (pure neuritis, or any number of skin lesions and neuritis); or “the” demonstrated presence of bacilli in a slit-skin smear, irrespective of the number of lesions. [Row # 56-60]

-          BT cases can be included in the PB group, “while” BB, BL, and LL patients can be classified under the MB group. This chronic infectious disease is curable, “and disability can be prevented” if early treatment is provided. [Row # 61-63]

-          The WHO guidelines for the diagnosis, treatment, and prevention of leprosy are followed in Ecuador. [Row # 65-66]

-          Since the recommendation of MDT in 1981 by the WHO, “the” worldwide [Row # 68]

-          However, despite the decrease in case notification rates, “it has been suggested by” epidemiological and spatiotemporal studies done in Brazil and China “that” leprosy has focal distribution with potential for hidden cases, “necessitating” intensified targeted control activities to reach the goal of “the” ‘Towards Zero Leprosy’ strategy [Row # 73-77]

-          In a study conducted in the Philippines in 2013, static, low transmission of leprosy in endemic rural areas and among children was identified despite good MDT coverage, and a new approach of chemoprophylaxis/immunoprophylaxis was suggested to target leprosy elimination [Row # 77-80]

-          demographic factors such as “being male” and “older age” are the [Row # 81-82]

-          In 2020, the highest proportion of cases was detected in the Southeast Asian region, followed by the Americas, with “74% of all worldwide new cases reported by” Brazil, India, and Indonesia. A new case detection rate of 18.8 per million population was recorded in the Americas, which is higher than the worldwide new case detection rate of 16.4 per million population. [Row # 83-87]

-          “Still, in the last 10 years,” there have been [Row # 92-93]

-          A leprosy notification system called SIVE-ALERTA is used by the Ecuadorian Ministry of Public Health (MSP) through an electronic PAHO-supported platform called ViEpi (Vigilancia Epidemiológica) [Row # 99-101]

-          Currently, new cases are reported through the SIVE-ALERTA system using both passive and active notification strategies [Row # 103-104]

-          Focal areas with transmission and vulnerable populations will also be identified, which can be specifically targeted to achieve the goal of zero autochthonous cases in line with WHO's “Towards Zero Leprosy’ strategy” [Row # 107-109]

2.       Materials and Methods

-          Distinct ecological diversity is found in each region [Row # 117].

-          In recent years, the MSP passive and active notification strategies “were used” to report new cases through the SIVE-ALERTA system 35. [Row # 148-149]

-          PB leprosy cases “were characterized by” 1-5 skin lesions, without the demonstrated presence of bacilli in the skin smear. MB cases “were characterized by over 5 skin lesions; nerve involvement; or the demonstrated presence of bacilli in a slit-skin smear, irrespective of the number of lesions [Row # 163-168]

-          Descriptive statistics “such as” frequency (%) for categorical variables and median [Row # 190]

3.       Results

-          “One hundred” and four of the cases (86.7%) were detected in the Sierra region [Row # 268-269]

-          The provinces with the highest number of cases of leprosy during the study period were Guayas (44.8%), Los Rios (15.7%), Pichincha (13.1%), El Oro (4.7%), and Loja (4.4%) [Row # 337-338]

4.       Discussion

-          Brazilian studies have “suggested” that the higher burden … [Row # 436]

-          The appearance of MB cases in Brazil was “directly proportional” to increased age, due to the long incubation period of leprosy, combined with a late diagnosis [Row # 442-443]

-          As immune-senescence occurs with aging, more severe clinical presentations are observed in the elderly population, which “sometimes go” unnoticed .. [Row # 448-449]

-          Cases of indeterminate leprosy, tuberculoid leprosy (TT), and borderline-tuberculoid leprosy (BT) were grouped in the paucibacillary (PB) group, “while” borderline-borderline (BB), borderline-lepromatous (BL), and lepromatous (LL) cases were classified under the multibacillary (MB) group [Row # 455-458]

-          Additionally, Santo Domingo de los Tsachilas province (located “on the” Coast) [Row # 481-482]

-          “Therefore”, the case notification rate might not be representative “of” the incidence, and the infection trends might be over or underestimated [Row # 540-541]

Comments 2: Methods 1. The authors should add the description about the four ecological regions. It is informative for potential reviewers of this manuscript.

Response 2: We agree with your observations. We have revised the text and added the following text to the section 2.1. Study Area: [“The Costa (Pacific Coast) consists of fertile tropical plains, hills, and low-elevation regions located in the western portion of the country, including the Pacific coastline. Guayaquil (Ecuador’s largest city and main port city), Esmeraldas, and Manta are some examples of major cities located in the warm and humid Costa. The Sierra (Andes) region is in the central belt of Ecuador, extending itself from north to south, including the elevated Andes mountains and the volcano chain. This region is well known for its agricultural capacities, its national parks, and major cities, such as Quito (the country’s capital), Cuenca, and Riobamba. The Amazonian region extends from the eastern slopes of the Andes into the plains of the Amazonian forests. This region includes biosphere reserves and indigenous middle-size towns, such as Tena, Coca, and Lago Agrio. The Galapagos Islands are also known as the “Archipiélago de Colón” and it consists of 13 main volcanic islands and 17 islets, located about 1000 kilometers west of the mainland Pacific Coast”]. These changes can be found in Rows #: 118-130. Additionally, a short description of the four ecological regions was provided in the Title of Figure 1.

Comments 3: Methods 2. The reviewer acknowledges that the authors used official leprosy case data, but the accuracy of this data is crucial. The authors should provide further details regarding its accuracy.

Response 3: Thank you for your observation. We agree with your comment, and we added a paragraph describing the data sources and their accuracy in section 2.2. Study Subjects: To maintain the most reliable leprosy incidence report nationwide, incident cases from two official sources of the Ecuadorian government (INEC and SIVE-ALERTA) were obtained and analyzed. INEC datasets include the National Archive of Data and Statistical Metadata (Archivo Nacional de Datos y Metadatos Estadísticos - ANDA) of the Ecuadorian National Institute of Statistics and Censuses which registry all deaths and hospital discharges (including leprosy cases) reported at the national level by the General Direction of Civil Registry and the Ecuadorian Ministry of Health. INEC provides the official government website for ANDA, available at https://anda.inec.gob.ec/anda/index.php/catalog. SIVE-ALERTA (Epidemiological-Surveillance-System) collects cases through surveillance of outpatients with leprosy diagnosis attending a health unit with a later confirmation using laboratory testing or by epidemiological linkage” [Row #: 149-160].

Comments 4: Methods 3. Does the surveillance system adequately cover all leprosy cases?

Response 4: We agree with your concern. We have revised the text and added the following text to the section 2.2. Study Subjects:Leprosy is a national notifiable disease in Ecuador, and all health personnel (public or private) are obligated to report suspected and confirmed cases. SIVE.ALERTA is part of the “Dirección Nacional de Vigilancia Epidemiológica” (National Directorate of Epidemiological Surveillance). The Epidemiological Gazette provides timely national information generated by the operational establishments of the Public Health System and Complementary Networks. This information is collected from the SIVE-ALERTA surveillance subsystem, which monitors events with high epidemic potential, outbreaks, and endemic diseases (publicly available at https://www.salud.gob.ec/gaceta-indicadores-2024/). Therefore, all suspected and posteriorly confirmed leprosy cases are registered in SIVE-ALERTA (a subsystem of the Ministry of Public Health of Ecuador) and later in INEC. In recent years, the MSP passive and active notification strategies were used to report new cases through the SIVE-ALERTA system”. [Row #: 137-149].

Comments 5: Methods 4 Were there any changes in the reporting system during the study period? If so, the authors should describe them. This could also be a limitation of the study.

Response 5: Thank you for your observation. We added text to the “Discussion” section: This can be explained by the changes in the reporting systems of leprosy throughout the study period. For instance, the histological classification of leprosy was most widely used during the first two years of the study period. Posteriorly, the Ecuadorian health authorities adopted the recommended clinical classification by PAHO/WHO (Paucibacilalry/Multibacillary). Nationally, both classifications are used and recognized. However, the microscopic classification is more efficiently implemented in urban areas with access to clinical laboratory diagnosis, while the clinical classification by PAHO/WHO is more practical in rural areas with poor access to laboratory services. Regardless of the clinical classification used by the Ecuadorian Ministry of Public Health (MSP), it is important to recognize that Ecuador is a developing country with 36% of its population living in remote rural areas. In these scenarios, the availability of microscopy is limited; therefore, the microscopic differentiation between the tuberculoid-lepromatous spectrum is impossible. Additionally, technological advances also played a role in the reporting strategies in Ecuador. From the year 2000-2012, obligatory notifiable diseases (such as leprosy) were reported manually through the submission of the form EPI-2 by healthcare workers. In 2013 the MSP implemented the online “Integrated System of Epidemiological Surveillance” (Sistema Integrado de Vigilancia Epidemiológica - SIVE) and its health emergency notification system called SIVE-ALERTA. Modernizing the reporting system improved the efforts to efficiently detect and control infectious diseases such as leprosy. [Row #: 459-479].

Comments 6: Results. The authors should adjust Figure 3 to make the numbers easier to read.

Response 6: Thank you for pointing this out. We modified the image by increasing the font size of the numbers and bolding the Incidence Rate numbers and the number of cases, placed inside the bars of the Figure. A short description was added in the caption of the figure: [“The incidence rates were placed above the bars and the number of cases was placed inside the bars of the figure”]. [Row #: 289-290].

Comments 7: Discussion 1. The authors should provide a more detailed discussion of the regions/provinces with a high number of leprosy cases.

Response 7: Thank you for your observation. We added text to the “Discussion” section: During the entire study period, the provinces with the highest number of leprosy cases were Guayas (44.8%), Los Rios (15.7%), Pichincha (13.1%), El Oro (4.7%), and Loja (4.4%); three of these provinces are located in the Costa region and the remaining two are located in the Sierra region. Historically, leprosy cases in Ecuador have been concentrated in Provinces and rural cantons from the Costa region. [Row #: 410-414].

Comments 8: Discussion 2. Why were 73% of the reported cases unspecified in this study?

Response 8: Thank you for your observation. We added text to the “Discussion” section to explain why 73% of the cases were classified as unspecified: “This can be explained by the changes in the reporting systems of leprosy throughout the study period. For instance, the histological classification of leprosy was most widely used during the first two years of the study period. Posteriorly, the Ecuadorian health authorities adopted the recommended clinical classification by PAHO/WHO (Paucibacilalry/Multibacillary). Nationally, both classifications are used and recognized. However, the microscopic classification is more efficiently implemented in urban areas with access to clinical laboratory diagnosis, while the clinical classification by PAHO/WHO is more practical in rural areas with poor access to laboratory services. Regardless of the clinical classification used by the Ecuadorian Ministry of Public Health (MSP), it is important to recognize that Ecuador is a developing country with 36% of its population living in remote rural areas. In these scenarios, the availability of microscopy is limited; therefore, the microscopic differentiation between the tuberculoid-lepromatous spectrum is impossible. Additionally, technological advances also played a role in the reporting strategies in Ecuador. From the year 2000-2012, obligatory notifiable diseases (such as leprosy) were reported manually through the submission of the form EPI-2 by healthcare workers. In 2013 the MSP implemented the online “Integrated System of Epidemiological Surveillance” (Sistema Integrado de Vigilancia Epidemiológica - SIVE) and its health emergency notification system called SIVE-ALERTA. Modernizing the reporting system improved the efforts to efficiently detect and control infectious diseases such as leprosy”. [Row #: 465-485].

Comments 9: Discussion 3. The authors should provide more detailed descriptions of 'urbanization and other forms of migration' in Ecuador.

Response 9: Thank you for pointing this out. We agree with your suggestion and we added text to explain this point in the “Discussion” section: During the entire study period, the provinces with the highest number of leprosy cases were Guayas (44.8%), Los Rios (15.7%), Pichincha (13.1%), El Oro (4.7%), and Loja (4.4%); three of these provinces are located in the Costa region and the remaining two are located in the Sierra region. Historically, leprosy cases in Ecuador have been concentrated in Provinces and rural cantons from the Costa region. Ecuador's significant urbanization began in the first half of the 20th century. This trend continues, with large cities including Guayaquil and Quito, absorbing a large portion of the rural population over the last decades. People moving from rural to urban areas may bring previously undiagnosed cases into cities, where they are more likely to be detected due to better medical care. This can contribute to fluctuations in case numbers as rural regions may have lower detection capacities. Moreover, in areas of Ecuador with higher numbers of immigrants from neighbouring countries, leprosy cases may rise if migrants come from Andean regions with higher endemicity status (E.g. Venezuela, Colombia, and Bolivia). Ecuador's border with Colombia and Peru, for instance, could be focal points for this type of migration-related fluctuation [Row #: 410-427].

Comments 10: Discussion 4. The authors should include a concluding statement for this study.

Response 10: Thank you for pointing this out. We agree with this comment. Therefore, we have added a “5. Conclusion” section: “We conclude that Ecuador has effectively applied global guidelines for the control and elimination of leprosy in the past twenty-three years, as a strong downward trend was observed during the study period. However, few cases remain in some cantons from the Pacific Coast and the Sierra region of the country. Therefore, it is imperative to maintain surveillance, prevention, and control strategies in those cantons with higher risk of leprosy detection and the most vulnerable populations, such as male adults over 40 years from the Costa and Sierra regions (such as cantons from El Oro, Santo Domingo de los Tsachilas, Los Rios and Imbabura provinces). While tangible progress is being made, continued efforts are needed to halt the transmission of leprosy in Ecuador and ultimately reach the goal of leprosy elimination”. [Row #: 551-560].

[Here, mention any other clarifications you would like to provide to the journal editor/reviewer.]

Reviewer 3 Report

Comments and Suggestions for Authors

This manuscript could be more short to stimulated more lectors, but I am not epidemiologist to avaible.

I think be need compareted with endemic countries as Brasil because is very stranger that the most cases are Paucibacilar, no specific type and hospitalized. I suggest article of Barreto Josafa.

Author Response

Comments 1: I think be need to compare with endemic countries as Brazil because is very strange that the most cases are Paucibacilar, have no specific type, and are hospitalized. I suggest the article of Barreto Josafa.

Response 1: Thank you for pointing this out. We agree with this comment. Therefore, we have added text to our “Discussion” section: However, it is important to consider that underdiagnosis of leprosy in children under 15 years of age is common in developing countries, due to the wide variety of clinical presentations and difficulty in performing in-situ clinical peripheral nerve evaluation. Therefore, it is imperative to strengthen clinical diagnosis capabilities to diagnose all cases of leprosy, including atypical cases in pediatric populations. [Row # 402-406]

Round 2

Reviewer 1 Report

Comments and Suggestions for Authors

Thank you for addressing my comments